# Simplified Action Decoder for Deep Multi-Agent Reinforcement Learning

**Hengyuan Hu, Jakob N Foerster**
Facebook AI Research, CA, USA
{hengyuan,jnf}@fb.com

## Abstract

In recent years we have seen fast progress on a number of benchmark problems in AI, with modern methods achieving near or super human performance in Go, Poker and Dota. One common aspect of all of these challenges is that they are by design *adversarial* or, technically speaking, zero-sum. In contrast to these settings, success in the real world commonly requires humans to collaborate and communicate with others, in settings that are, at least partially, cooperative. In the last year, the card game *Hanabi* has been established as a new benchmark environment for AI to fill this gap. In particular, Hanabi is interesting to humans since it is entirely focused on *theory of mind*, *i.e.*, the ability to effectively reason over the intentions, beliefs and point of view of other agents when observing their actions. Learning to be informative when observed by others is an interesting challenge for Reinforcement Learning (RL): Fundamentally, RL requires agents to explore in order to discover good policies. However, when done naively, this randomness will inherently make their actions less informative to others during training. We present a new deep multi-agent RL method, the *Simplified Action Decoder* (SAD), which resolves this contradiction exploiting the centralized training phase. During training SAD allows other agents to not only observe the (*exploratory*) action chosen, but agents instead also observe the *greedy* action of their team mates. By combining this simple intuition with best practices for multi-agent learning, SAD establishes a new SOTA for learning methods for 2-5 players on the self-play part of the Hanabi challenge. Our ablations show the contributions of SAD compared with the best practice components. All of our code and trained agents are available at https://github.com/facebookresearch/Hanabi_SAD.

## 1 Introduction

Humans are highly social creatures and spend vast amounts of time coordinating, collaborating and communicating with others. In contrast to these, at least partially, cooperative settings most progress on AI in games has been in zero-sum games where agents compete against each other, typically rendering communication futile. This includes examples such as Go (Silver et al., 2016; 2017; 2018), poker (Brown & Sandholm, 2017; Moravčík et al., 2017; Brown & Sandholm, 2019) and chess (Campbell et al., 2002).

This narrow focus is unfortunate, since communication and coordination require unique abilities. In order to enable smooth and efficient social interactions of groups of people, it is commonly required to reason over the intents, points of views and beliefs of other agents from observing their actions. For example, a driver can reasonably infer that if a truck in front of them is slowing down when approaching an intersection, then there is likely an obstacle ahead. Furthermore, humans are both able to interpret the actions of others and can act in a way that is informative when their actions are being observed by others, capabilities that are commonly called *theory of Mind* (ToM), (Baker et al., 2017). Importantly, in order to carry out this kind of reasoning, an agent needs to consider why a given action is taken and what this decision indicates about the state of the world. Simply observing what other agents are doing is not sufficient.

While these abilities are particularly relevant in partially observable, fully cooperative multi-agent settings, ToM reasoning clearly matters in a variety of real world scenarios. For example, autonomous

cars will likely need to understand the point of view, intents and beliefs of other traffic participants in order to deal with highly interactive settings such as 4-way crossing or dense traffic in cities.

Hanabi is a fully cooperative, partially-observable card game that has recently been proposed as a new benchmark challenge problem for AI research (Bard et al., 2019) to fill the gap around ToM. In Hanabi, players need to find conventions that allow them to effectively exchange information from their local observations through their actions, taking advantage of the fact that actions are observed by all team mates.

Most prior state-of-the-art agents for Hanabi were developed using handcrafted algorithms, which beat off-the-shelf deep multi-agent RL methods by a large margin. This makes intuitive sense: Beyond the "standard" multi-agent challenges of credit assignment, nonstationarity and joint exploration, learning an informative policy presents an additional fundamentally new conflict. On the one hand, an RL agent needs to explore in order to discover good policies through trial and error. On the other hand, when carried out naively, this exploration will add noise to the policy of the agent during the training process, making their actions strictly less informative to their team mates.

One possible solution to this is to explore in the space of deterministic partial policies, rather than actions, and sample these policies from a distribution that conditions on a *common knowledge* Bayesian belief. This is successfully carried out in the Bayesian Action Decoder (BAD) (Foerster et al., 2019), the only previous Deep RL method to achieve a state-of-the-art in Hanabi. While this is a notable accomplishment, it comes at the cost of simplicity and generality. For a start, BAD requires an explicit common knowledge Bayesian belief to be tracked, which not only adds computational burden due to the required sampling steps, but also uses expert knowledge regarding the game dynamics. Furthermore, BAD, as presented, is trained using actor-critic methods which are sample inefficient and suffer from local optima. In order to get around this, BAD uses population based training, further increasing the number of samples required. Lastly, BAD's explicit reliance on *common knowledge* limits the generality of the method.

In this paper we propose the *Simplified Action Decoder* (SAD), a method that achieves a similar goal to BAD, but addresses all of the issues mentioned above. At the core of SAD is a different approach towards resolving the conflict between exploration and being interpretable, which, like BAD, relies on the *centralized training* with *decentralized control* (CT/DC) regime. Under CT/DC information can be exchanged freely amongst all agents during *centralized training*, as long as the final policies are compatible with *decentralized execution*.

The key insight is that during training we do not have to chose between being informative, by taking greedy actions, and exploring, by taking random actions. To be informative, the greedy actions do not need to be executed by the environment, but only need to be observed by the team mates. Thus in SAD each agent takes two different actions at each time step: One greedy action, which is not presented to the environment but observed by the team mates at the next time step as an additional input, and the "standard" (exploratory) action that gets executed by the environment and is observed by the team mates as part of the environment dynamics. Importantly, during greedy execution the observed environment action can be used instead of centralized information for the additional input, since now the agent has stopped exploring.

Furthermore, to ensure that these greedy actions and observations get decoded into a meaningful representation, we can optionally train an auxiliary task that predicts key hidden game properties from the action-observation trajectories. While we note that this idea is in principle compatible with any kind of model-free deep RL method with minimal modifications to the core algorithm, we use a distributed version of recurrent DQN in order to improve sample efficiency, account for partial observability and reduce the risk of local optima. We also train a joint-action Q-function that consists of the sum of per-agent Q-values to allow for off-policy learning in this multi-agent setting using Value Decomposition Networks (VDN) (Sunehag et al., 2017).

Using SAD we establish a new SOTA for learning methods for 2-5 players in Hanabi, with a method that not only requires less expert knowledge and compute, but is also more general than previous approaches. In order to ensure that our results can be easily verified and extended, we also evaluate our method on a proof-of-principle matrix game and open-source our training code and agents. Beyond enabling more research into the self-play aspect of Hanabi, we believe these resources will provide a much needed starting point for the ad-hoc teamwork part of the Hanabi challenge.

## 2    RELATED WORK

Our work relates closely to research on emergent communication protocols using deep multi-agent RL, as first undertaken by Sukhbaatar et al. (2016) and Foerster et al. (2016) . There has been a large number of follow-up papers in this area, so listing all relevant work is beyond the scope and we refer the reader to Nguyen et al. (2018), a recent survey on deep multi-agent RL. One major difference to our work is that the environments considered typically contain a cheap-talk channel, which can be modeled as a continuous variable during the course of training. This allows agents to, for example, use differentiation across the communication channel in order to learn protocols. In contrast, in our setting agents have to communicate through the observable environment actions themselves, requiring fundamentally different methods.

Furthermore, our work is an example of cooperative multi-agent learning in partially observable settings under centralized training and decentralized control. There have been a large number of papers in this space, with seminal work including MADDPG (Lowe et al., 2017) and COMA (Foerster et al., 2018a), both of which are actor-critic methods that employ a centralized critic with decentralized actors. Again, we refer the reader to Nguyen et al. (2018) for a more comprehensive survey.

Until 2018, work on Hanabi had been focused on hand-coded methods and heuristics. Some relevant examples include SmartBot (O'Dwyer, 2019) and the so-called "hat-coding" strategies, as implemented by WTFWThat (Wu, 2018). These strategies use the information theoretic ideas that allow each hint to reveal information to all other agents at the same time. While they do not perform well for 2-player Hanabi due to the smaller action space, they get near perfect scores for 3-5 players.

In contrast, so far learning methods have seen limited success on Hanabi. Bard et al. (2019) undertake a systematic evaluation of current Deep RL methods for 2-5 players in two different regimes and open-source the Hanabi-Learning-Environment (HLE) to foster research on the game. They evaluate a feed-forward version of DQN trained on 100 million samples and a recurrent actor-critic agent with population based training using 20 billion samples. Notably, while both agents achieve near 0% win rate for 3-5 players in Hanabim, at a high level their DQN agent is a good starting point for our work. However, since the authors did not propose any specific method of accounting for the issues introduced by $\epsilon$-greedy exploration in a ToM task, they resorted to setting $\epsilon$ to zero after a short burn-in phase. The only state-of-the-art in Hanabi established by an RL agent is from Foerster et al. (2019) which we refer to in more detail in Section 1 and Section 4. Recently there have also been attempts to train agents that are robust to different team-mates (Canaan et al., 2019) and even to extend to human-AI collaboration (Liang et al., 2019). For a more comprehensive review on previous results on Hanabi we refer the reader to Bard et al. (2019).

Poker is another partially observable multi-agent setting, although it is fundamentally different due to the game being zero-sum. Recent success in Poker has extensively benefited from search (Brown et al.). Examples of using search in Hanabi include Goodman (2019).

## 3    BACKGROUND

### 3.1    SETTING

In this paper we assume a Dec-POMDP (Oliehoek, 2012), in which $N$ agents interact in a partially observable environment. At each time step agent $a \in 1..N$ obtains an observation, $o_t^a = O(s_t, a)$, where $s_t \in \mathcal{S}$ is the Markov state of the system and $O(s_t, a)$ is the deterministic observation function. Since we are interested in ToM, in our setting the observation function includes the last action of the acting agent, which is observed by all other agents at the next time step. We note that actions are commonly observable not only in board games but also in some real world multi-agent settings, such as autonomous driving.

For simplicity, we restrict ourselves to turn based settings, in which at each time step only the acting agents takes an action, $u_t^a$, which is sampled from their policy, $u^a \sim \pi_\theta^a(u^a|\tau^a)$, while all other agents take a no-op action. Here $\tau^a$ is the action-observation history of agent $a$, $\tau^a = \{o_0^a, u_0^a, r_1, ..r_T, o_T^a\}$, $T$ is the length of the episode and $\theta$ are the weights of a function approximator that represents the policy, in our case recurrent neural networks, such as LSTMs (Hochreiter & Schmidhuber, 1997).

We further use $\tau_t$ to describe the state-action sequence, $\tau = \{s_0, \mathbf{u_0}, r_1, ..r_T, s_T\}$, where $\mathbf{u_t}$ is the joint action of all agents.

As is typical in cooperative multi-agent RL, the goal of the agents is to maximize the total expected return, $J_\theta = \mathbb{E}_{\tau \sim P(\tau|\theta)} R_0(\tau)$, where $R_0(\tau)$ is the return of the trajectory (in general $R_t(\tau) = \sum_{t' \geq t} \gamma^{t'-t} r_{t'}$) and $\gamma$ is an optional discount factor. We have also assumed that agents are sharing parameters, $\theta$, as is common in cooperative MARL.

## 3.2 Distributed Recurrent DQN and Auxiliary Tasks

In Q-learning the agent approximates the expected return for a given state action-pair, $s, u$, assuming that the agent acts greedily with respect to the Q-function for all future time steps, $Q(s, u) = \mathbb{E}_{\tau \sim P(\tau|s,u)} R_t(\tau)$, where $\tau = \{s_t, u_t, r_{t+1}, \ldots, s_T\}$, $u_t = u$ and $u_{t'} = \arg\max_{u'} Q(s_{t'}, u'), \forall t' > t$. A common exploration scheme is $\epsilon$-greedy, in which the agent takes a random action with probability $\epsilon$ and acts greedily otherwise. Importantly, the Q-function can be trained efficiently using the Bellman equation: $Q(s, u) = \mathbb{E}_{s'}[r_{t+1} + \gamma \max_{u'} Q(s', u')]$, where for simplicity we have assumed a deterministic reward. In Deep Q-Learning (DQN) (Mnih et al., 2015) the Q-function is parameterized by a deep neural network and trained with transitions sampled from experience replay.

In our work we also incorporate other best practice components of the last few years, including double-DQN (van Hasselt et al., 2015), dueling network architecture (Wang et al., 2015) and prioritized replay (Schaul et al., 2015). We also employ a distributed training architecture similar to the one proposed by Horgan et al. (2018) where a number of different actors with their own exploration rates collect experiences in parallel and feed them into a central replay buffer. Since our setting is partially observable the natural choice for the function approximator is a recurrent neural network. A combination of these techniques was first explored by Kapturowski et al. (2019) in single agent environments such as Atari and DMLab-30.

Another common best-practice in RL are auxiliary tasks Mirowski et al. (2016); Jaderberg et al. (2016), in which the agent produces extra output-heads that are trained on supervised tasks and optimized alongside the RL loss.

## 3.3 Centralised Training, Decentralized Execution and Joint Q-functions

The most straight forward application of Q-learning to multi-agent settings is Independent Q-Learning (IQL) (Tan, 1993) in which each agent keeps an independent estimate of the expected return, treating all other agents as part of the environment. One challenge with IQL is that the exploratory behavior of other agents is not corrected for via the $\max$ operator in the bootstrap. Notably, IQL does typically not take any advantage of *centralized training* with *decentralised control* (CT/DC), a paradigm under which information can be exchanged freely amongst agents during the training phase as long as the policies rely only on local observations during execution.

There are various approaches for learning joint-Q-functions in the CT/DC regime. For example, Value-Decomposition-Networks (VDN) (Sunehag et al., 2017) represent the joint-Q-function as a sum of per-agent contributions and QMIX (Rashid et al., 2018) learns a non-linear but monotonic combination of these contributions.

# 4 Method

## 4.1 Theory of Mind and Bayesian Reasoning

At the very core of interpreting the actions of another agent, and ToM in general, is Bayesian reasoning. Fundamentally, asking *what* a given action by another agent implies about the *state of the world* requires understanding of *why* this action was taken. To illustrate this, we start out with an agent that has a given belief about the state-action history of the world, $\tau_t$, given her own action-observation history $\tau_t^a$: $B(\tau_t) = P(\tau_t|\tau_t^a)$.

Next the agent observes the action $u_t^{a'}$ of her team mate, $a'$, and carries out a Bayesian update:

$$P(\tau_t | \tau_t^a, u_t^{a'}) = \frac{P(u_t^{a'} | \tau_t) P(\tau_t | \tau_t^a)}{\sum_{\tau_t'} P(u_t^{a'} | \tau_t') P(\tau_t' | \tau_t^a)} \tag{1}$$

$$= \frac{\pi^{a'}(u_t^{a'} | O(a', \tau_t)) B(\tau_t)}{\sum_{\tau_t'} \pi^{a'}(u_t^{a'} | O(a', \tau_t')) B(\tau_t')}, \tag{2}$$

where, with a slight abuse of notation, we have used (and will keep using) $O(a', \tau_t)$ for the action-observation history, $\tau_t^{a'}$, that results from applying the observation function for agent $a'$ to $\tau_t$ at each time step. Note that for non-deterministic observation functions we would have to marginalize over $P(\tau_t^{a'} | \tau_t)$.

Clearly, since agents have access to the policy of their teammate during centralised training, we could in principle evaluate this *explicit* Bayesian belief. However, beyond the practical difficulty of computing this *explicit* belief, when it is used as an input to the policy it will lead to prohibitively costly higher order beliefs. The typical workout for this is a *public belief* over *private features* which only conditions on common knowledge and can therefore be calculated by all agents individually, we refer to Moravčík et al. (2017); Nayyar et al. (2013); Foerster et al. (2018b) for more details.

Instead, in this work we rely on RNNs to learn *implicit* representations of the sufficient statistics over the distribution of the Markov state given the action-observation histories, noting that they are unlikely to recover exact beliefs due to the issues mentioned above.

## 4.2 EXPLORATION AND BELIEFS

Next we illustrate the impact of exploration on the beliefs, which we will do in the explicit (exact) case, since it serves as an upper bound on the accuracy of the implicit beliefs. Since we are looking at fully-cooperative settings we assume that the optimal policy of the agent is deterministic and any randomness is due to exploration. Given that we are focused on value based methods we furthermore assume an $\epsilon$-greedy exploration scheme, noting that the same analysis can be extended to other methods. Under this exploration scheme $\pi^{a'}(u_t^{a'} | O(a', \tau_t))$ becomes:

$$\pi^{a'}(u_t^{a'} | O(a', \tau_t)) = (1 - \epsilon) \mathbf{I}(u^*(\tau_t), u_t^{a'}) + \epsilon / |U|, \tag{3}$$

where we have used $u^*(\tau_t)$ to indicate the greedy action of the agent $a'$, $u^*(\tau_t) = \arg\max_u Q^{a'}(u, O(a', \tau_t))$ and $\mathbf{I}$ is the indicator function.

While the first part corresponds to a filtering operator, in which the indicator function only attributes finite probability to those histories that are consistent with the action taken under greedy execution, the exploration term adds a fixed (history independent) probability, which effectively 'blurs' the posterior:

$$P(\tau_t | \tau_t^a, u_t^{a'}) = \frac{\left((1 - \epsilon) \mathbf{I}(u^*(\tau_t), u_t^{a'}) + \epsilon / |U|\right) B(\tau_t)}{\sum_{\tau'} \left((1 - \epsilon) \mathbf{I}(u^*(\tau'), u_t^{a'}) + \epsilon / |U|\right) B(\tau')} \tag{4}$$

$$= \frac{\left((1 - \epsilon) \mathbf{I}(u^*(\tau_t), u_t^{a'}) + \epsilon / |U|\right) B(\tau_t)}{\epsilon / |U| + \sum_{\tau'} \left((1 - \epsilon) \mathbf{I}(u^*(\tau'), u_t^{a'})\right) B(\tau')} \tag{5}$$

$$= \frac{B(\tau_t)}{1 + |U| \sum_{s'} \left((1/\epsilon - 1) \mathbf{I}(u^*(\tau'), u_t^{a'}) B(\tau')\right)} \tag{6}$$

$$+ \frac{\left((1 - \epsilon) \mathbf{I}(u^*(\tau_t), u_t^{a'})\right) B(\tau_t)}{\epsilon / |U| + \sum_{\tau'} \left((1 - \epsilon) \mathbf{I}(u^*(\tau'), u_t^{a'})\right) B(\tau')}. \tag{7}$$

We find that the posterior includes an additional term of the form $B(\tau_t)$ which carries over an *unfiltered* density over the trajectories from the prior. We further confirm that in the limit of $\epsilon = 1$, the posterior collapses to the prior, $P(\tau_t | \tau_t^a, u_t^{a'}) = B(\tau_t)$. This can be particularly worrisome in

the context of our training setup, whereby different agents run different, and potentially high, $\epsilon$ throughout the course of training. It fundamentally makes the beliefs obtained less informative.

While not making the above argument explicitly, the Bayesian Action Decoder (BAD) (Foerster et al., 2019), resolves this issue by shifting exploration to the level of deterministic partial policies, rather than action-level, and tracking an approximate Bayesian belief. As outlined in Section 1 this comes at a huge cost in the complexity of the method, the computation requirements and in the loss of generality of the method.

### 4.3 SIMPLIFIED ACTION DECODING

In this paper we take a drastically simpler and different approach towards the issue. We note that the 'blurring', which makes decoding of an action challenging, is entirely due to the $\epsilon$-greedy exploration term. Furthermore, in order for another agent to do an implicit Bayesian update over an action taken, it is not required that this action is executed by the environment. Indeed, if we assume that other agents can observe the *greedy* action, $u^*$, at every time step and condition their belief update on this, the terms depending on $\epsilon$ disappear from the Bayesian update:

$$P(\tau_t|\tau_t^a, u^*) = \frac{\mathbf{I}(u^*(\tau_t), u^*) B(\tau_t)}{\sum_{\tau'} \mathbf{I}(u^*(\tau'), u^*) B(\tau')} \tag{8}$$

Therefore, to have our cake *and* eat it, in the Simplified Action Decoder (SAD) the acting agent is allowed to 'take' two actions at any given time step during training. The first action, $u^a$, is the standard environment action, which gets executed as usual and is observed by all agents through the observation function at the next time step, as mentioned in Section 3. The second action, $u^*$, is the greedy action of the active agent. This action does not get executed by the environment but instead is presented as an additional input to the other agents at the next time step, taking advantage of the centralized training regime during which information can be exchanged freely.

Clearly we are not allowed to pass around extra information during *decentralized control*, but luckily this is not needed. Since we set $\epsilon$ to 0 at test time we can simply use the, now greedy, environment action obtained from the observation function as our greedy-action input.

While this is most straight forward in settings where the last action is observed by other agents directly, in principle SAD can also be extended to settings where it is indirectly observed by all agents through the environment dynamics. In these cases we can replace the greedy-action side-channel with a learned inverse model that recovers the action from the observation history during execution.

Furthermore, to encourage the agent to meaningfully decode the information contained in the greedy action, we can optionally add an auxiliary task to the training process, such as predicting unobserved information from their observation history .

While this idea is compatible with any deep RL algorithm with minimal modifications, we use a recurrent version of DQN with distributed training, dueling networks and prioritized replay. We also learn a joint Q-function using VDN in order to address the challenges of multi-agent off-policy learning, please see Section 3 for details on all of these standard methods.

## 5 EXPERIMENTS

### 5.1 MATRIX GAME

We first verify the effectiveness of SAD in the two step, two player matrix game from Foerster et al. (2019), which replicates the *communication through action* challenge of Hanabi in a highly simplified setting. In this fully cooperative game each player obtains a privately observed 'card', which is drawn iid from two options (1,2).

After observing her card, the first player takes one of three possible discrete actions (1, 2, 3). Crucially, the second player observes both her own private card and the team mate's action before acting herself, which establishes the opportunity to communicate. The payout is a function of both the two private cards and the two actions taken by both agents, as shown in Figure 1.

Importantly, there are some obvious strategies that do not require any communication. For example, if both player learn to play the 2nd action, the payout is always 8 points, independent of the cards dealt. However, if the players do learn to communicate it is possible to achieve 10 points for every pair of cards dealt.

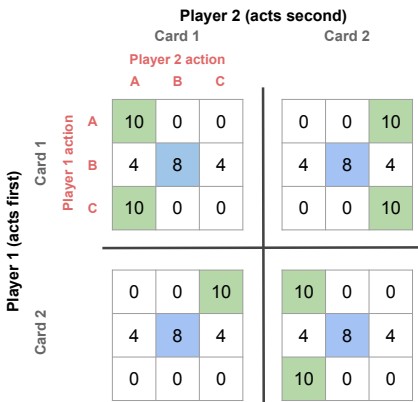

## 5.2 HANABI

Hanabi is a fully cooperative card game in which all players work together to complete piles of cards referred to as *fireworks*. Each card has a rank, **1** to **5**, and a color, **G / B / W / Y / R**. Each firework (one per color) starts with a **1** and is finished once the **5** has been added. There are three **1**s, one **5** and two of all other ranks for each of the colors, adding up to a total of 50 cards in the deck. The twist in Hanabi is that while players can observe the cards held by their team mates, they cannot observe their own cards and thus need to exchange information with each other in order to understand what cards can be played.

Figure 1: Illustration of the matrix game from Foerster et al. (2019)

There are two main means for doing so: First of all, players can take grounded *hint actions*, in which they reveal the subset of a team mate's hand that matches a specific rank or color. An example hint is "Your third and fifth card are **1**s". These hint actions cost scarce *information tokens*, which can be replenished by *discarding* a card, an action that both removes the card from the game and makes it visible to all players.

Finally players can also choose to *play* a card. If this card is the next card for the firework of the corresponding color, it is added to the firework and the team scores one point. Otherwise the card is removed from the game, the identity is made public, and the team loses one of the 3 life tokens. If the team runs out of life tokens before the end of the game, all points collected so far are lost and the game finishes immediately. These rules result in a maximum score of $5 \times 5 = 25$ points in any game, which corresponds to all five fireworks being completed with five cards per firework.

To ensure reproducibility and comparability of our results we use the Hanabi Learning Environment (HLE) (Bard et al., 2019) for all experimentation. For further details regarding Hanabi and the self-play part of the Hanabi challenge please see Bard et al. (2019).

## 5.3 ARCHITECTURE AND COMPUTATION REQUIREMENTS

We borrow some ideas and insights from prior distributed Q-learning methods while bring extensions to MARL as well as innovations to improve throughput and efficiency. Following Horgan et al. (2018) and Kapturowski et al. (2019), we use a distributed prioritized replay buffer shared by $N$ asynchronous actors and a centralized trainer that samples mini-batches from the replay buffer to update the model. In each actor thread, we run $K$ environments sequentially and batch their observations together. The observation batch is then fed into an actor that utilize a GPU to compute a batch of actions. All asynchronous actors share one GPU and the trainer uses another GPU for gradient computation and model updates. This is different from prior works which run single actor and single environment in each thread on a CPU. Our method enables us to run a very large number of simulations with moderate computation resources. In all Hanabi experiments, we run $N = 80$ actor threads with $K = 80$ environments in each thread on single machine with 40 CPU cores and 2 GPUs. Without this architectural improvement, it may require at least a few hundred CPU cores to run 6400 Hanabi environments, in which case neural network agents and simulations have to be distributed across multiple machines, greatly reducing the reproducibility and accessibility of such research. Please refer to Appendix A for implementation details and hyper-parameters.

## 6 RESULTS

### 6.1 MATRIX GAME

As we can see in Figure 2, even in our simple matrix game the *greedy action input* makes a drastic difference. With an average reward of around 9.5 points, tabular IQL does well in this task, matching the *BAD* results from Foerster et al. (2019). However, just by adding the greedy action as an additional input, we obtain an average performance of $9.97 \pm 0.02$. Results are averaged over 100 seeds, and shading is s.e.m. The code is available here: www.bit.ly/2mBJLyk.

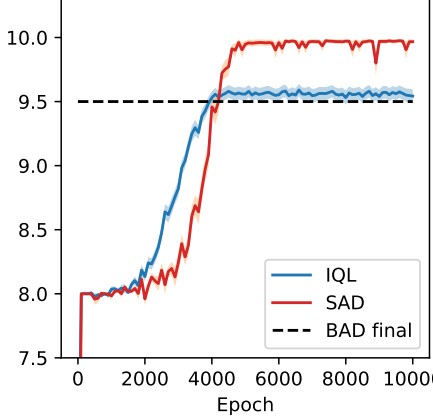

Figure 2: Results for the matrix game.

### 6.2 HANABI

As shown in Table 1, our findings from the matrix game are for the most part confirmed on the challenging Hanabi benchmark. To illustrate the contributions of the different components, we compare average scores and win rates across 13 independent training runs of SAD and three different options: *IQL* is simply the recurrent DQN agent with parameter sharing, *VDN* is the same agent but also learns a joint Q-function and finally *SAD & AuxTask* is the SAD agent with the auxiliary task.

While we find that SAD significantly outperforms our baselines (IQL and VDN) for 2, 4 and 5 players in terms of average score and/or win rate, there is no significant difference for 3 players, where VDN matches the performance of SAD.

Interestingly, the auxiliary task only significantly helps the 2-player performance, where it substantially boosts the average score and win rate. In contrast, it drastically hurts performance for 3-5 players, which opens an interesting avenue for future work.

For completeness we have included training curves showing average scores and s.e.m. across all training runs for all numbers of players for our methods and ablations in Appendix B. We find that for 5 players the auxiliary task drastically reduces the variance of SAD and intermittently leads to higher performance during training but ultimately results in lower final performance. We can also clearly see that despite 72 hours of training and billions of samples consumed, the performance has not plateaued for 3-5 players, pointing to an obvious avenue for further improvements.

The original numbers in the Hanabi challenge and BAD used population based training (Jaderberg et al., 2018), effectively reporting maximum performance across a large number of different runs. Therefore, for reproducibility purposes, we report evaluations of the best model from our various training runs for each method in Table 2.

As shown, under this reporting we establish a new SOTA for learning methods on the self-play part of the Hanabi challenge for 2-5 players, with the most drastic improvements being achieved for 3-5 players. In particular, we beat both the ACHA agent from Bard et al. (2019) and the BAD agent on average score, even though both of them used population based training and require more compute. We note that while we follow the counting convention proposed by the challenge paper, BAD was optimized for a different counting scheme, in which agents keep their scores when they run out of lives. This may explain the higher win rate (58.6%) of BAD combined with a relatively low mean score, which is exceeded even by our baseline methods. Once again, only the performance for 2-player is significantly improved by the auxiliary task and the 3-player setting is an outlier in the sense that SAD does not improve the best performance compared to VDN.

## 7 CONCLUSION AND FUTURE WORK

In this paper we presented the Simplified Action Decoder (SAD), a novel deep multi-agent RL algorithm that allows agents to learn communication protocols in settings where no cheap-talk channel is available. On the challenging benchmark Hanabi our work substantially improves the SOTA for an RL method for all numbers of players. For two players SAD establishes a new high-score

| Agent | 2 Players | 3 Players | 4 Players | 5 Players |
|---|---|---|---|---|
| IQL (Baseline) | $23.77 \pm 0.04$ $43.88 \pm 1.21\%$ | $23.02 \pm 0.10$ $26.16 \pm 2.01\%$ | $21.99 \pm 0.09$ $10.15 \pm 0.86\%$ | $20.60 \pm 0.11$ $2.27 \pm 0.32\%$ |
| VDN (Baseline) | $23.83 \pm 0.03$ $44.97 \pm 1.28\%$ | $\mathbf{23.71 \pm 0.06}$ $41.16 \pm 1.27\%$ | $23.03 \pm 0.15$ $23.57 \pm 2.20\%$ | $21.18 \pm 0.12$ $2.26 \pm 0.32\%$ |
| SAD | $23.87 \pm 0.03$ $47.90 \pm 1.10\%$ | $\mathbf{23.69 \pm 0.05}$ $41.12 \pm 1.10\%$ | $\mathbf{23.27 \pm 0.16}$ $29.38 \pm 2.63\%$ | $\mathbf{22.06 \pm 0.23}$ $7.22 \pm 1.29\%$ |
| SAD AuxTask | $\mathbf{24.02 \pm 0.01}$ $54.38 \pm 0.41\%$ | $23.56 \pm 0.07$ $40.77 \pm 1.35\%$ | $22.78 \pm 0.10$ $20.80 \pm 1.65\%$ | $21.47 \pm 0.08$ $2.92 \pm 0.40\%$ |

Table 1: Mean performance of our methods and baselines on Hanabi. We take the final models of 13 independent runs, i.e. 13 models per algorithm per player setting. Each model is evaluated on 100K games. Mean and s.e.m over the mean scores of the 13 models are shown in the table. The second row of each section is the win rate.

| Agent | 2 Players | 3 Players | 4 Players | 5 Players |
|---|---|---|---|---|
| Rainbow (Bard et al., 2019) | $20.64 \pm 0.03$ $2.5\%$ | $18.71 \pm 0.01$ $0.2\%$ | $18.00 \pm 0.17$ $0\%$ | $15.26 \pm 0.18$ $0\%$ |
| ACHA (Bard et al., 2019) | $22.73 \pm 0.12$ $15.1\%$ | $20.24 \pm 0.15$ $1.1\%$ | $21.57 \pm 0.12$ $2.4\%$ | $16.80 \pm 0.13$ $0\%$ |
| BAD (Foerster et al., 2019) | $23.92 \pm 0.01$ $58.56\%$ | - - | - - | - - |
| IQL (Baseline) 50.47% | $23.97 \pm 0.01$ $40.25\%$ | $23.69 \pm 0.01$ $19.39\%$ | $22.76 \pm 0.01$ $4.93\%$ | $21.29 \pm 0.01$ |
| VDN (Baseline) | $23.96 \pm 0.01$ $50.27\%$ | $\mathbf{23.99 \pm 0.01}$ $50.37\%$ | $23.79 \pm 0.00$ $38.86\%$ | $21.80 \pm 0.01$ $4.98\%$ |
| SAD | $24.01 \pm 0.01$ $52.39\%$ | $23.93 \pm 0.01$ $48.05\%$ | $\mathbf{23.81 \pm 0.01}$ $41.45\%$ | $\mathbf{23.01 \pm 0.01}$ $13.93\%$ |
| SAD & AuxTask | $\mathbf{24.08 \pm 0.01}$ $56.09\%$ | $23.81 \pm 0.01$ $49.74\%$ | $23.47 \pm 0.01$ $33.87\%$ | $22.25 \pm 0.01$ $7.33\%$ |

Table 2: Comparison between the previous SOTA learning methods and ours. We take the **best** model of 13 runs for each of our methods and baselines. Each model is evaluated on 100K games with different seeds. Mean and s.e.m over the **100K** games are shown in the table. The s.e.m. is less than 0.01 for most models. Bold numbers are the best results achieved with learning algorithms. The second row of each section is the win rate.

across any method. Furthermore we accomplish all of this with a method that is both simpler and requires less compute than previous advances. While these are encouraging steps, there is clearly more work to do. In particular, there remains a large performance gap between the numbers achieved by SAD and the known performance of *hat-coding* strategies (Wu, 2018) for 3-5 players. One possible reason is that SAD does not undertake any explicit exploration in the space of possible conventions. Another promising route for future work is to integrate search with RL, since this has produced SOTA results in a number of different domains including Poker, Go and backgammon.

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

# A    NETWORK ARCHITECTURE AND HYPER-PAMAMETERS FOR HANABI

Our Hanabi agent uses dueling network architecture (Wang et al., 2015). The main body of the network consists of 1 fully connected layer of 512 units and 2 LSTM (Hochreiter & Schmidhuber, 1997) layers of 512 units, followed by two output heads for value and advantages respectively. The same network configuration is used across all Hanabi experiments. We take the default featurization of HLE and replace the card knowledge section with the V0-Belief proposed by Foerster et al. (2019). The maximum length of an episode is capped at 80 steps and the entire episode is stored in the replay buffer as one training sample. This avoids the "slate hidden states" problem as described in Kapturowski et al. (2019) because we can simply initialize the hidden states of LSTM as zero during training. For exploration and experience prioritization, we follow the simple strategy as in Horgan et al. (2018) and Kapturowski et al. (2019). Each actor executes an $\epsilon_i$-greedy policy where $\epsilon_i = \epsilon^{1+\frac{1}{N-1}\alpha}$ for $i \in \{0, ..., N-1\}$ but with a smaller $\epsilon = 0.1$ and $\alpha = 7$. For simplicity, all players of a game use the same epsilon. The per time-step priority $\delta_t$ is the TD error and per episode priority is computed following $\delta_e = \eta \max_t \delta_i + (1 - \eta)\hat{\delta}$ where $\eta = 0.9$. Priority exponent is set to 0.9 and importance sampling exponent is set to 0.6. We use $n$-step return (Sutton, 1988) and double Q-learning (van Hasselt et al., 2015) for target computation during training. The discount factor $\gamma$ is set to 0.999. The network is updated using Adam optimizer (Kingma & Ba, 2014) with learning rate $lr = 6.25 \times 10^{-5}$ and $\epsilon = 1.5 \times 10^{-5}$. Trainer sends its network weights to all actors every 10 updates and target network is synchronized with online network every 2500 updates. These hyper-parameters are fixed across all experiments.

In the baseline, we use Independent Q-Learning where each player estimates the Q value and selects action independently at each time-step. Note that all players need to operate on the observations in order to update their recurrent hidden states while only the current player has non-trivial legal moves and other players can only select 'pass'. Each player then writes its own version of the episode into the prioritized replay buffer and they are sampled independently during training. The prioritized replay buffer contains $2^{17}(131072)$ episodes. We warm up the replay buffer with 10,000 episodes before training starts. Batch size during training is 128 for games of different numbers of players.

As mentioned in Section 4, the SAD agent is built on top of joint Q-function where the Q value is the sum of the individual Q value of all players given their own actions. One episode produces only one training sample with an extra dimension for the number of players. The replay buffer size is reduced to $2^{16}$ for 2-player and 3-player games and $2^{15}$ for 4-player and 5-player games. The batch sizes for 2-, 3-, 4-, 5-players are 64, 43, 32, 26 respectively to account for the fact that each sample contains more data.

Auxiliary task can be added to the agent to help it decode the greedy action more effectively. In Hanabi, the natural choice is the predict the card of player's own hand. In our experiments, the auxiliary task is to predict the status of a card, which can be playable, discardable, or unknown. The loss is the average cross entropy loss per card and is simply added to the TD-error of reinforcement learning during training.

# B  LEARNING CURVES FOR HANABI

Figure 3 shows learning curves of different algorithms averaged over 13 seeds per algorithm per player setting. Shading is error of the mean.

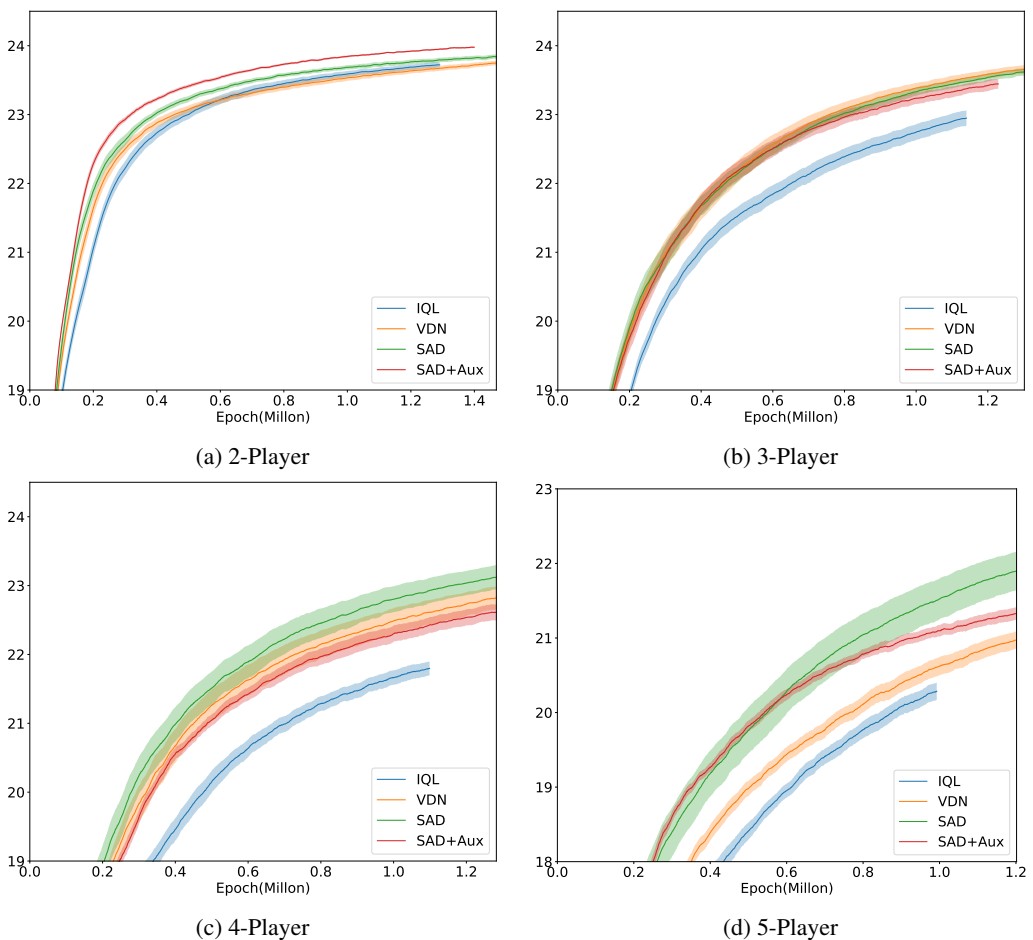

(a) 2-Player

(b) 3-Player

(c) 4-Player

(d) 5-Player

Figure 3: Learning Curves

