# OpenReview forum: "Simplified Action Decoder for Deep Multi-Agent Reinforcement Learning"
_ICLR.cc/2020/Conference — Accept (Spotlight)_

### Official Review · AnonReviewer2 · 2019-10-18
**Official Blind Review #2**

**Rating:** 8

**Review:**

This paper introduces a novel exploitation of the centralized training for decentralized execution regime for Dec-POMDPs with publicly/commonly known actions.  In particular, the paper augments independent value-based reinforcement learning by allowing each agent to announce the action it would have taken, had it acted greedily. This relaxation is consistent with decentralized execution, at which time agents always act greedily and no counterfactual announcement is required. The paper demonstrates the utility of this trick in Hanabi, where it achieves strong performance when combined with distributed Q-learning and an auxiliary task. The paper is largely well-written.

I think that the trick introduced in this paper is a valuable contribution to the community. As the paper discusses, SAD is simpler and easier to implement than BAD, the algorithm with the most comparable ambitions with respect to scalability. Also, unlike BAD, it is model-free. I think that investigating lightweight alternatives to common/public knowledge based approaches, such as SAD, is an important research direction.

That being said, I have some issues with the presentation of the content in the paper.

1. Section 4 is problematic. The entirety of the section is based on the assumption that agent a’ observes the Markov state. The paper claims that this assumption is made for simplicity: “where for simplicity we have assumed that agent a’ uses a feed-forward policy and observes the Markov state of the environment.”
    a. First, this assumption is unmet, with few exceptions. Other agents do
        NOT in general observe the Markov state. This is a very significant
        part of why Dec-POMDPs are so difficult. Justifying SAD in a setting
        that so radically departs from its use case is not informative.
    b. Second, the claim made by the paper (that the assumption is made
        for simplicity) is misleading. The assumption is made out of
        necessity. The cost of neglecting public/common knowledge, as SAD
        does, is that principled Bayesian reasoning is not possible. It is
        important that the paper acknowledges this to counterbalance its
        (valid) criticisms that public/common knowledge based approaches
        are difficult to scale.
2. That the experiments are expensive is understandable and a fully complete ablation study is not expected. But the results of these experiments should be presented clearly.
    a. In both Table 1 and Table 2, presenting the “best seed” is
        not appropriate. The paper should report the mean or the median.
        Given that only three seeds could be afforded, it is understandable
        that the results would have high variance.
    b. In Table 2, (if I am understanding the paper correctly) it is claimed
        that the s.e.m. is less than or equal to 0.01 over a set of three seeds
        for all of the experiments. But looking at Figure 4, this is clearly not
        the case, especially in the four and five player settings. The intended
        meaning should be clarified.
    c. In Table 1, the paper should report the results for the baseline, VDN,
        VDN with greedy input, and SAD separately, as is done in Table 2.
        These are the main experimental results and merit space in the main
        body of the paper. Moreover, there are a number of results that
        deserve written attention.
        i. It looks the baseline is very competitive with BAD in two player
           Hanabi. This in itself is a very interesting finding and is worth
           discussing.
        ii. Looking at Table 2, it appears that VDN with GreedyInput
            outperforms SAD in three of the four settings and quite
            significantly in the five player setting. If this is also the case when
            results are aggregated over the median or mean, it should be
            discussed.
3. The greedy input approach is specific to Dec-POMDPs in which actions are publicly/commonly available. This should at least be mentioned somewhere.

Overall, I think the ideas and experimental findings in the paper are very interesting. However, as outlined above, there are a number of issues. At a high level, I think that the paper is too concerned with 1) justifying greedy input with Bayesian reasoning and 2) promoting state of the art results. The ideas and experimental findings are more than sufficient for strong contribution without these things.

In its current state, I feel that this work is a rejection. However, its issues are relatively easily amendable:
1. For each algorithm and each setting, separately report median or mean results over the seeds, with uncertainty.
2. Reallocate space to section 6.2 for a scientific discussion of the experimental results.
3. Remove or qualify the arguments made in section 4.
4. Mention that the greedy input approach is specific to Dec-POMDPs with publicly/commonly available actions.

Were these issues addressed, my opinion would change.

**Experience Assessment:**

I have read many papers in this area.

**Review Assessment: Checking Correctness Of Derivations And Theory:**

I carefully checked the derivations and theory.

**Review Assessment: Checking Correctness Of Experiments:**

I assessed the sensibility of the experiments.

**Review Assessment: Thoroughness In Paper Reading:**

I read the paper thoroughly.

---

> ### Author Response · Authors · 2019-11-07
> **Response**
>
> First of all, many thanks for an extremely thorough and insightful review. Below we address each of the individual requests.
>
> @1)
> As suggested by R2, we have redone the derivations in Section 4. We now account for partial observability throughout. We had originally chosen to use an agent with access to the Markov state as a simplified setting to motivate the method, but as pointed out by R2 this does hide interesting technical and conceptual challenges.
>
> We also added a paragraph that explains the issues around explicit Bayesian beliefs and the rationale for using common knowledge beliefs instead. We believe that in the current formulation the motivation is appropriate, since implicit Bayesian beliefs are not affected by the same kind of recursive explosion.
>
> @2)
> To make these numbers meaningful we carried out further experimentations resulting in 13 seeds for each of the settings and algorithms. While we do not believe that there is a meaningful way to report averages and uncertainties across the original 3 seeds, we now report average results and best results (incl. uncertainties) for each of the settings and algorithms in the updated Table 1 and updated Table 2. We have updated the captions of both tables to clarify how the numbers were computed.
>
> We also include training curves showing the average performance of our methods (including s.e.m.) across the training process.
>
> @2.b)
> The numbers in Table 2 are obtained by first select the best model among 13 runs of each algorithm and then evaluating that model over 100K games with different seeds. The original numbers in the Hanabi challenge and BAD used population based training, effectively taking a max over 32 runs. Therefore, we do believe it is important for reproducibility purposes to also report the results for our best models.
>
> @2.c)
> We have also added a more in depth scientific discussion of the results in Section 6, as suggested by R2.
> In particular we point out that the auxiliary task only helps for 2 players and VDN matches the performance of SAD for 3 players.
>
> @3)
> Indeed, SAD requires all agents to be able to observe the last actions. This was mentioned in the later part of Section 3.1 (“. Since we are interested in ToM, in our setting the observation function includes the last action of the acting agent, which is observed by all other agents at the next time step. We note that actions are commonly observable not only in board games but also in some real world multi-agent settings, such as autonomous driving.”). We now moved this towards the beginning of Section 3.1 to emphasize the assumption.
>
> We also added a paragraph to Section 4.3 which explores how the greedy-action input could be extended to settings where the last action is not directly observed, but can in principle be decoded by all agents through the environment dynamics and observation function.
>
> Other questions:
> @"VDN with GreedyInput outperforms SAD":
> Yes - the auxiliary task only helps for the 2 player setting. As mentioned in our ‘general comments’ we have now restructured the paper to clearly mark the auxiliary task as optional, rather than being part of the core method. We also believe that it is an interesting question for future work to investigate why the auxiliary task only helps for 2 players.
>
> We added some discussion on this point to Section 6.2.

---

> > ### Comment · AnonReviewer2 · 2019-11-09
> > **Re: Response**
> >
> > Thanks for your thoughtful response. Some further comments are provided below.
> >
> > @1) I have a small gripe with the statement "implicit Bayesian beliefs are not affected by the same kind of recursive explosion." It seems to me that implicit beliefs are still affected by the same kind of recursive explosion. SAD avoids this by learning a representation that retains much of the useful information but does not necessarily correspond to any particular Bayesian belief (and thereby is not really an implicit Bayesian belief). In any case, I see what you are saying and think that the revised version of section 4 motivates the method nicely.
> >
> > @2.b) That is a fair point. I had not considered population based training from that perspective. Accordingly, I think that including scores for both the mean and the best seed, as is done in the revised version, makes the most sense.
> >
> > Overall, the revised version addresses the concerns I had in my original review. I have revised my score correspondingly.

---

> > > ### Author Response · Authors · 2019-11-11
> > > **Thank you for the timely response to our revision.**
> > >
> > > @”It seems to me that implicit beliefs are still affected by the same kind of recursive explosion. SAD avoids this by learning a representation that retains much of the useful information but does not necessarily correspond to any particular Bayesian belief (and thereby is not really an implicit Bayesian belief)."
> > >
> > > Re: Yes, this is a good point. We will see if there is a good way to get this idea across and update the paper accordingly for the next version. There is also an interesting question for future work here, to investigate what kind of belief SAD actually learns. It might turn out to be something close to a Bayesian belief, with the caveats around recursive reasoning (though at this point we simply do not know).
> > >
> > > @”@2.b) .. I had not considered population based training from that perspective. Accordingly, I think that including scores for both the mean and the best seed, as is done in the revised version, makes the most sense.”
> > >
> > > Yes - this aspect of PBT is not normally mentioned and it does make a fair comparison with other methods more challenging.
> > >
> > > @Overall, the revised version addresses the concerns I had in my original review. I have revised my score correspondingly.
> > >
> > > Perfect, thanks again for the extremely constructive comments and the fast turn around. This is a great example of the review process working well. I wish it was more common.

---

### Official Review · AnonReviewer3 · 2019-10-19
**Official Blind Review #3**

**Rating:** 8

**Review:**

The paper examines the problem of epsilon-greedy exploration in cooperative multi-agent reinforcement learning. Such exploration makes actions less informative to other agents because of the noise added to greedy optimal actions. The suggested solution is to consider two actions: one action is epsilon-greedy and it is passed to the environment, while another is fully greedy according to the agent’s strategy, and it is shown to another agents. At test time these two actions are the same. This idea is applied to Hanabi game; the self-play regime is examined. The suggested method is claimed to show state-of-art results in 2-5 players Hanabi.

The Methods Section provides mathematical grounds of using Bayesian reasoning in ToM. It is shown how epsilon-greedy exploration leads to the blurring of the posterior and consequently making beliefs of other agents less informative. Thus, additional fully greedy input is motivated. However, there are still some vague parts in the description of methods used.
1.	In Section 4.3 it is not clear what auxiliary supervised task is added to training. The information from Appendix should be moved to Section 4.3. Furthermore, some details on how training data for this task is collected and model is trained should be added. Also, how does this model affect the whole training procedure?
2.	The description of the Q-learning in the first paragraph of Section 3.2 is too vague. In the first equation for Q(s,u), there is no u_t, u_{t’}, so it’s unclear why they pop out next. In the Bellman equation for Q(s, u) parentheses are omitted.

The advantage of the paper is that apart from the suggested additional greedy actions, several ingredients of the state-of-art approach (suggested in previous works) are tested separately, such as learning a joint Q-function and auxiliary supervised task. However, considering the results, it seems there is no clear winner for different number of players in Hanabi. From the Table 2, it is not clear whether the additional greedy input or the auxiliary task are beneficial for the best seed results. One of the three competing methods is the method named VDN in Tables 1 and 2. Is it the contribution of the paper or a previous work? If it is the contribution, the difference from the previous work should be clearly stated. If not, then it’s not fully fair to say that the new approach shows state-of-art results.

Also, is it fair to say that s.e.m. of the best seed is 0.01? Comparing the results from Figure 3 and Table 2, one can see that the order of the winners is changing significantly, so 3 seeds are not enough to reliably estimate the performance of the best seed and s.e.m. should be higher.

Another comment on the Experiment 6.1: as far as BAD results are mentioned in the text, maybe they could be put on the plot? At least final training score as horizontal line, if the whole training curve is not available.

Minor comments
1.	Page 1: Simply observing other agents are doing -> Simply observing what other agents are doing.
2.	Page 3: While our method are general, we restrict ourselves to turn based settings – odd phrase.
3.	Page 4: A combination these techniques -> A combination of these techniques.
4.	\eps-greedy is written with a hyphen in some places of the paper and somewhere without.
5.	Section 5.3: Compute Requirements -> Computation Requirements.

UPD: score updated.


**Experience Assessment:**

I do not know much about this area.

**Review Assessment: Checking Correctness Of Derivations And Theory:**

I assessed the sensibility of the derivations and theory.

**Review Assessment: Checking Correctness Of Experiments:**

I assessed the sensibility of the experiments.

**Review Assessment: Thoroughness In Paper Reading:**

I read the paper at least twice and used my best judgement in assessing the paper.

---

> ### Author Response · Authors · 2019-11-07
> **Response**
>
> We thank the reviewer for an insightful review.
>
> @Auxiliary task in Section 4.3:
> The intention behind SAD is to learn an implicit representation over the sufficient statistics given the trajectory. We explore the impact of an auxiliary loss function during training that encourages the neural network to learn the appropriate representations. You can think of the auxiliary task as a supervised loss term that gets optimized on top of the RL loss. Auxiliary tasks are fairly standard in deep RL, but we have added further details to the background section of the paper.
>
> @"The description of Q-learning.. too vague":
> We have clarified the notation in this section. However, we also believe that Q-learning is a relatively well established method and expect that most readers will be broadly familiar with it. Furthermore, understanding Q-learning is not strictly required for understanding the rest of the paper. It simply happens to be the underlying learning algorithm of choice in our paper.
>
> @"VDN & other ingredients of the methods":
> We do not claim VDN as part of our contribution. Notably we introduce VDN in the background section where we cite the original paper and mention it under ‘best practice’ in other parts of the paper.
>
> To further clarify, we have updated Table 1 and Table 2, where we compare the mean scores and best scores of different ablations. These tables now include a “Baseline” label under IQL and VND. These tables not only clearly illustrate the effectiveness of the key idea of SAD, but also provide some insights into the relative improvements due to VDN, SAD and the auxiliary task.
>
> We have also added more discussion in Section 6.2.
>
> @“s.e.m. of the best seed is 0.01”:
> Yes, this is the uncertainty over true average score of our best model of our best run when evaluated for 100K games not over 3 runs. Also, as suggested by Reviewer #2, we have added a new table (Table 1 in the updated version) to show the mean and s.e.m. of our method & ablations over 13 different training runs.
>
> @”Experiment 6.1.. BAD results”:
> We have now added a dashed line to indicate the final mean performance of BAD and will include actual training curves in the CRC. Note that for clarity and simplicity our version of the matrix game uses a tabular representation rather than  deep NN. Using tabular learning, SAD achieves perfect scores and even IQL exceeds the performance of BAD from the original paper (which used function approximation).

---

> > ### Author Response · Authors · 2019-11-11
> > **Further comments / questions?**
> >
> > @Reviewer #3:
> > As mentioned above we have addressed the comments and concerns in the new version of the paper (uploaded on the 5th of Nov). Please let us know if there are any further questions. Many thanks!

---

> > ### Comment · AnonReviewer3 · 2019-11-14
> > **Re: Response**
> >
> > Thanks for the fruitful comments and explanations. The presentation of the results with mean and max of 13 seeds seems much more solid. The performance gain solely due to greedy output is now clearly demonstrated.  I have updated the score.

---

> > > ### Author Response · Authors · 2019-11-14
> > > **Re^2: Response**
> > >
> > > Once again, many thanks for the detailed feedback and also for the fast response.

---

### Official Review · AnonReviewer1 · 2019-10-25
**Official Blind Review #1**

**Rating:** 8

**Review:**

The paper presents SAD (Simplified Action Decoder), a new method to address an issue in Centralized Training / Decentralized Control regimes: that exploratory actions during training can make those actions less informative for the agents who observe and attempt to learn from it. The method addresses this issue by allowing the agent to select two actions: one (the possibly exploratory action) is applied to the environment and the other (the greedy action) is presented to other agents as part of their observations for the next turn.

The authors use a distributed recurrent DQN architecture and apply the resulting agent to a toy problem and to the Hanabi Learning Environment. The authors claim that the method offers improvement over the Bayesian Action Decoder (BAD), that has been similarly applied to the same environments, being simpler, more sample efficient  and achieving overall higher scores, which is confirmed by their results: the agent outperforms BAD in 2-player Hanabi (where BAD was previously state-of-the-art) and the best scores out of any learning agent (although not as high as some non-learning agents such as WTFWThat in the 3-5 player versions.

The paper does not directly address the ad-hoc cooperation aspect of Hanabi, and it is unclear wheter the method could be used as-is for that problem, due to its reliance on centralized training. Nevertheless, the paper represents a relevant improvement to the self-play aspect of the game, and the core insight that the method leverages could conceivably be applied to minimize the noise introduced by exploration in other cooperative CT/DC problems. For this reason, I recommend the paper to be accepted.

Typos/language issues:

Introduction: “spend vast amounts of time coordinate” -> coordinating
Section 3.1: “While our method are general” -> methods

"accomplish a state of the art in Hanabi"
I see what you mean but this is a strange phrasing.


**Experience Assessment:**

I have published one or two papers in this area.

**Review Assessment: Checking Correctness Of Derivations And Theory:**

I did not assess the derivations or theory.

**Review Assessment: Checking Correctness Of Experiments:**

I assessed the sensibility of the experiments.

**Review Assessment: Thoroughness In Paper Reading:**

I read the paper at least twice and used my best judgement in assessing the paper.

---

> ### Author Response · Authors · 2019-11-07
> **Response**
>
> Many thanks for the encouraging feedback. Indeed, we are focussed on the self-play part of the challenge. Ad-hoc teamwork is an exciting future direction which could potentially also benefit from this kind of method.
>
> For example it is imaginable that a large population of interesting policies could be trained using SAD, which would be a great starting point for ad-hoc teamplay.
>
> As mentioned in the paper, we plan to open source the code and agents of our paper which hopefully will kick-start the investigation of ad-hoc teamplay in Hanabi.

---

### Author Response · Authors · 2019-11-07
**Overall response**

We would like to thank all reviewers for the insightful reviews and comments.

One fairly consistent concern amongst the reviewers was the presentation of the results and the lack of clarity regarding the key contribution of the method compared to existing best practice.

We have taken this to heart and improved the presentation of the results with clearer focus on the novel aspect of SAD, i.e. the observation of the greedy-action during training. As requested by R2, we now also present mean scores including s.e.m. along with our best-in-class results across an increased number of runs (13 seeds). These results show that on average SAD significantly outperforms our ablations for 2,4 and 5 players and matches the performance of VDN for 3 players. We have also added a more granular discussion of the results to section 6.2

To clarify that the auxiliary task is optional compared to the core idea, we now call this combination “SAD & AuxTask” across the paper.

---

### Decision · Program_Chairs · 2019-12-19

**Decision:**

Accept (Spotlight)

**Comment:**

The method presented, the simplified action decoder, is a clever way of addressing the influence of exploratory actions in multi-agent RL. It's shown to enable state of the art performance in Hanabi, an interesting and relatively novel cooperative AI challenge. It seems, however, that the method has wider applicability than that.

All reviewers agree that this is good and interesting work. Reviewer 2 had some issues with the presentation of the results and certain assumptions, but the authors responded so as to alleviate any concerns.

This paper should definitely be accepted, if possible as oral.